# DIFFUSION POLICIES AS AN EXPRESSIVE POLICY CLASS FOR OFFLINE REINFORCEMENT LEARNING

**Zhendong Wang**[1,*] , **Jonathan J Hunt**[2,†] , **Mingyuan Zhou**[1,†]
[1]The University of Texas at Austin, [2] Twitter
`zhendong.wang@utexas.edu, jhunt@twitter.com`
`mingyuan.zhou@mccombs.utexas.edu`

## ABSTRACT

Offline reinforcement learning (RL), which aims to learn an optimal policy using a previously collected static dataset, is an important paradigm of RL. Standard RL methods often perform poorly in this regime due to the function approximation errors on out-of-distribution actions. While a variety of regularization methods have been proposed to mitigate this issue, they are often constrained by policy classes with limited expressiveness that can lead to highly suboptimal solutions. In this paper, we propose representing the policy as a diffusion model, a recent class of highly-expressive deep generative models. We introduce Diffusion Q-learning (Diffusion-QL) that utilizes a conditional diffusion model to represent the policy. In our approach, we learn an action-value function and we add a term maximizing action-values into the training loss of the conditional diffusion model, which results in a loss that seeks optimal actions that are near the behavior policy. We show the expressiveness of the diffusion model-based policy, and the coupling of the behavior cloning and policy improvement under the diffusion model both contribute to the outstanding performance of Diffusion-QL. We illustrate the superiority of our method compared to prior works in a simple 2D bandit example with a multimodal behavior policy. We then show that our method can achieve state-of-the-art performance on the majority of the D4RL benchmark tasks.

## 1 INTRODUCTION

Offline reinforcement learning (RL), also known as batch RL, aims at learning effective policies entirely from previously collected data without interacting with the environment (Lange et al., 2012; Fujimoto et al., 2019). Eliminating the need for online interaction with the environment makes offline RL attractive for a wide array of real-world applications, such as autonomous driving and patient treatment planning, where real-world exploration with an untrained policy is risky, expensive, or time-consuming. Instead of relying on real-world exploration, offline RL emphasizes the use of prior data, such as human demonstration, that is often available at a much lower cost than online interactions. However, relying only on previously collected data makes offline RL a challenging task. Applying standard policy improvement approaches to an offline dataset typically leads to relying on evaluating actions that have not been seen in the dataset, and therefore their values are unlikely to be estimated accurately. For this reason, naive approaches to offline RL typically learn poor policies that prefer out-of-distribution actions whose values have been overestimated, resulting in unsatisfactory performance (Fujimoto et al., 2019).

Previous work on offline RL generally addressed this problem in one of four ways: 1) regularizing how far the policy can deviate from the behavior policy (Fujimoto et al., 2019; Fujimoto & Gu, 2021; Kumar et al., 2019; Wu et al., 2019; Nair et al., 2020; Lyu et al., 2022); 2) constraining the learned value function to assign low values to out-of-distribution actions (Kostrikov et al., 2021a; Kumar et al., 2020); 3) introducing model-based methods, which learn a model of the environment dynamics and perform pessimistic planning in the learned Markov decision process (MDP) (Kidambi

---

*The work was done in part during a summer internship at Twitter.
†Joint senior authors; order determined by flipping a coin.

et al., 2020; Yu et al., 2021); 4) treating offline RL as a problem of sequence prediction with return guidance (Chen et al., 2021; Janner et al., 2021; 2022). Our approach falls into the first category.

Empirically, the performance of policy-regularized offline RL methods is typically slightly worse than that of other approaches, and here we show that this is largely because the policy regularization methods perform poorly due to their limited ability to accurately represent the behavior policy. This results in the regularization adversely affecting the policy improvement. For example, the policy regularization may limit the exploration space of the agent to a small region with only suboptimal actions and then the Q-learning will be induced to converge towards a suboptimal policy.

The inaccurate policy regularization occurs for two main reasons: 1) policy classes are not expressive enough; 2) the regularization methods are improper. In most prior work, the policy is a Gaussian distribution with mean and diagonal covariance specified by the output of a neural network. However, as offline datasets are often collected by a mixture of policies, the true behavior policy may exhibit strong multi-modalities, skewness, or dependencies between different action dimensions, which cannot be well modeled by diagonal Gaussian policies (Shafiullah et al., 2022). In a particularly extreme, but not uncommon example, a Gaussian policy is used to fit bimodal training data by minimizing the Kullback–Leibler (KL) divergence from the data distribution to the policy distribution. This will result in the policy exhibiting mode-covering behavior and placing high density in the middle area of the two modes, which is actually the low-density region of the training data. In such cases, regularizing a new policy towards the behavior-cloned policy is likely to make the policy learning substantially worse. Second, the regularization, such as the KL divergence and maximum mean discrepancy (MMD) (Kumar et al., 2019), is often not well suited for offline RL. The KL divergence needs access to explicit density values and MMD needs multiple action samples at each state for optimization. These methods require an extra step by first learning a behavior cloned policy to provide density values for KL optimization or random action samples for MMD optimization. Regularizing the current policy towards the behavior cloned policy can further induce approximation errors, since the cloned behavior policy may not model the true behavior policy well, due to limitations in the expressiveness of the policy class. We conduct a simple bandit experiment in Section 4, which illustrates these issues can occur even on a simple bandit task.

In this work, we propose a method to perform policy regularization using diffusion (or score-based) models (Sohl-Dickstein et al., 2015; Song & Ermon, 2019; Ho et al., 2020). Specifically, we use a multilayer perceptron (MLP) based denoising diffusion probabilistic model (DDPM) (Ho et al., 2020) as our policy. We construct an objective for the diffusion loss which contains two terms: 1) a behavior-cloning term that encourages the diffusion model to sample actions in the same distribution as the training set, and 2) a policy improvement term that attempts to sample high-value actions (according to a learned Q-value). Our diffusion model is a conditional model with states as the condition and actions as the outputs. Applying a diffusion model here has several appealing properties. First, diffusion models are very expressive and can well capture multi-modal distributions. Second, the diffusion model loss constitutes a strong distribution matching technique and hence it could be seen as a powerful sample-based policy regularization method without the need for extra behavior cloning. Third, diffusion models perform generation via iterative refinement, and the guidance from maximizing the Q-value function can be added at each reverse diffusion step.

In summary, our contribution is Diffusion-QL, a new offline RL algorithm that leverages diffusion models to do precise policy regularization and successfully injects the Q-learning guidance into the reverse diffusion chain to seek optimal actions. We test Diffusion-QL on the D4RL benchmark tasks for offline RL and show this method outperforms prior methods on the majority of tasks. We also visualize the method on a simple bandit task to illustrate why it can outperform prior methods. Code is available at `https://github.com/Zhendong-Wang/Diffusion-Policies-for-Offline-RL`.

## 2 PRELIMINARIES AND RELATED WORK

**Offline RL.** The environment in RL is typically defined by a Markov decision process (MDP): $M = \{S, \mathcal{A}, P, R, \gamma, d_0\}$, with state space $S$, action space $\mathcal{A}$, environment dynamics $\mathcal{P}(s' \mid s, a) : S \times S \times \mathcal{A} \to [0, 1]$, reward function $R : S \times \mathcal{A} \to \mathbb{R}$, discount factor $\gamma \in [0, 1)$, and initial state distribution $d_0$ (Sutton & Barto, 2018). The goal is to learn policy $\pi_\theta(a \mid s)$, parameterized by $\theta$, that maximizes the cumulative discounted reward $\mathbb{E}\left[\sum_{t=0}^{\infty} \gamma^t r(s_t, a_t)\right]$. The action-value or Q-value of

a policy $\pi$ is defined as $Q^\pi(\boldsymbol{s}_t, \boldsymbol{a}_t) = \mathbb{E}_{\boldsymbol{a}_{t+1}, \boldsymbol{a}_{t+2}, \ldots \sim \pi} \left[ \sum_{t=0}^\infty \gamma^t r(\boldsymbol{s}_t, \boldsymbol{a}_t) \right]$. In the offline setting (Fu et al., 2020), instead of the environment, a static dataset $\mathcal{D} \triangleq \{(\boldsymbol{s}, \boldsymbol{a}, r, \boldsymbol{s}')\}$, collected by a behavior policy $\pi_b$, is provided. Offline RL algorithms learn a policy entirely from this static offline dataset $\mathcal{D}$, without online interactions with the environment.

**Diffusion Model.** Diffusion-based generative models (Ho et al., 2020; Sohl-Dickstein et al., 2015; Song & Ermon, 2019) assume $p_\theta(\boldsymbol{x}_0) := \int p_\theta(\boldsymbol{x}_{0:T}) d\boldsymbol{x}_{1:T}$, where $\boldsymbol{x}_1, \ldots, \boldsymbol{x}_T$ are latent variables of the same dimensionality as the data $\boldsymbol{x}_0 \sim p(\boldsymbol{x}_0)$. A forward diffusion chain gradually adds noise to the data $\boldsymbol{x}_0 \sim q(\boldsymbol{x}_0)$ in $T$ steps with a pre-defined variance schedule $\beta_i$, expressed as

$$q(\boldsymbol{x}_{1:T} \,|\, \boldsymbol{x}_0) := \textstyle\prod_{t=1}^T q(\boldsymbol{x}_t \,|\, \boldsymbol{x}_{t-1}), \quad q(\boldsymbol{x}_t \,|\, \boldsymbol{x}_{t-1}) := \mathcal{N}(\boldsymbol{x}_t; \sqrt{1 - \beta_t} \boldsymbol{x}_{t-1}, \beta_t \boldsymbol{I}).$$

A reverse diffusion chain, constructed as $p_\theta(\boldsymbol{x}_{0:T}) := \mathcal{N}(\boldsymbol{x}_T; \boldsymbol{0}, \boldsymbol{I}) \prod_{t=1}^T p_\theta(\boldsymbol{x}_{t-1} \,|\, \boldsymbol{x}_t)$, is then optimized by maximizing the evidence lower bound defined as $\mathbb{E}_q[\ln \frac{p_\theta(\boldsymbol{x}_{0:T})}{q(\boldsymbol{x}_{1:T} \,|\, \boldsymbol{x}_0)}]$ (Jordan et al., 1999; Blei et al., 2017). After training, sampling from the diffusion model consists of sampling $\boldsymbol{x}_T \sim p(\boldsymbol{x}_T)$ and running the reverse diffusion chain to go from $t = T$ to $t = 0$. Diffusion models can be straightforwardly extended to conditional models by conditioning $p_\theta(\boldsymbol{x}_{t-1} \,|\, \boldsymbol{x}_t, c)$.

**Related Work: Policy Regularization.** Most prior methods for offline RL in the class of regularized policies rely on behavior cloning for policy regularization: BCQ (Fujimoto et al., 2019) constructs the policy as a learnable and maximum-value-constrained deviation from a separately learned Conditional-VAE (CVAE, Sohn et al. (2015)) behavior-cloning model; BEAR (Kumar et al., 2019) adds a weighted behavior-cloning loss via minimizing MMD into the policy improvement step; TD3+BC (Fujimoto & Gu, 2021) applies the same trick as BEAR via maximum likelihood estimation (MLE); BRAC (Wu et al., 2019) evaluates multiple methods for behavior-cloning regularization, such as the KL divergence, MMD, and Wasserstein dual form; IQL (Kostrikov et al., 2021b) is an advantage weighted behavior-cloning method with "in-sample" learned Q-value functions. Goo & Niekum (2022) emphasize the necessity of conducting explicit behavioral cloning in offline RL, while Ajay et al. (2022) admit the power of conditional generative models for decision making.

**Related Work: Diffusion Models in RL.** Pearce et al. (2023) propose to better imitate human behaviors via diffusion models which are expressive and stable. Diffuser (Janner et al., 2022) applies a diffusion model as a trajectory generator. The full trajectory of state-action pairs form a single sample for the diffusion model. A separate return model is learned to predict the cumulative rewards of each trajectory sample. The guidance of the return model is then injected into the reverse sampling stage. This approach is similar to Decision Transformer (Chen et al., 2021), which also learns a trajectory generator through GPT2 (Radford et al., 2019) with the help of the true trajectory returns. When used online, sequence models can no longer predict actions from states autoregressively (since the states are an outcome of the environment). Thus, in the evaluation stage, a whole trajectory is predicted for each state while only the first action is applied, which incurs a large computational cost. Our approach employs diffusion models for RL in a distinct manner.

We apply the diffusion model to the action space and we form it as a conditional diffusion model with states as the condition. This approach is model-free and the diffusion model is sampling a single action at a time. Further, our Q-value function guidance is injected during training, which provides good empirical performance in our case. While both Diffuser (Janner et al., 2022) and our work apply diffusion models in Offline RL, Diffuser is from the model-based trajectory-planning perspective while our method is from the offline model-free policy-optimization perspective.

## 3 Diffusion Q-Learning

Below we explain how we apply a conditional diffusion model as an expressive policy for behavior cloning. Then, we introduce how we add Q-learning guidance into the learning of our diffusion model in the training stage with the behavior cloning term acting as a form of policy regularization.

### 3.1 Diffusion Policy

**Notation:** Since there are two different types of timesteps in this work, one for the diffusion process and one for reinforcement learning we use superscripts $i \in \{1, \ldots, N\}$ to denote diffusion timestep and subscripts $t \in \{1, \ldots, T\}$ to denote trajectory timestep.

We represent our RL policy via the reverse process of a conditional diffusion model as

$$\pi_\theta(\boldsymbol{a} \,|\, \boldsymbol{s}) = p_\theta(\boldsymbol{a}^{0:N} \,|\, \boldsymbol{s}) = \mathcal{N}(\boldsymbol{a}^N; \mathbf{0}, \boldsymbol{I}) \prod_{i=1}^{N} p_\theta(\boldsymbol{a}^{i-1} \,|\, \boldsymbol{a}^i, \boldsymbol{s})$$

where the end sample of the reverse chain, $\boldsymbol{a}^0$, is the action used for RL evaluation. Generally, $p_\theta(\boldsymbol{a}^{i-1} \,|\, \boldsymbol{a}^i, \boldsymbol{s})$ could be modeled as a Gaussian distribution $\mathcal{N}(\boldsymbol{a}^{i-1}; \boldsymbol{\mu}_\theta(\boldsymbol{a}^i, \boldsymbol{s}, i), \boldsymbol{\Sigma}_\theta(\boldsymbol{a}^i, \boldsymbol{s}, i))$. We follow Ho et al. (2020) to parameterize $p_\theta(\boldsymbol{a}^{i-1} \,|\, \boldsymbol{a}^i, \boldsymbol{s})$ as a noise prediction model with the covariance matrix fixed as $\boldsymbol{\Sigma}_\theta(\boldsymbol{a}^i, \boldsymbol{s}, i) = \beta_i \boldsymbol{I}$ and mean constructed as

$$\boldsymbol{\mu}_\theta(\boldsymbol{a}^i, \boldsymbol{s}, i) = \tfrac{1}{\sqrt{\alpha_i}} \big( \boldsymbol{a}^i - \tfrac{\beta_i}{\sqrt{1-\bar{\alpha}_i}} \boldsymbol{\epsilon}_\theta(\boldsymbol{a}^i, \boldsymbol{s}, i) \big).$$

We first sample $\boldsymbol{a}^N \sim \mathcal{N}(\mathbf{0}, \boldsymbol{I})$ and then from the reverse diffusion chain parameterized by $\theta$ as

$$\boldsymbol{a}^{i-1} \,|\, \boldsymbol{a}^i = \frac{\boldsymbol{a}^i}{\sqrt{\alpha_i}} - \frac{\beta_i}{\sqrt{\alpha_i(1-\bar{\alpha}_i)}} \boldsymbol{\epsilon}_\theta(\boldsymbol{a}^i, \boldsymbol{s}, i) + \sqrt{\beta_i}\boldsymbol{\epsilon}, \;\; \boldsymbol{\epsilon} \sim \mathcal{N}(\mathbf{0}, \boldsymbol{I}), \;\; \text{for } i = N, \dots, 1. \quad (1)$$

Following DDPM (Ho et al., 2020), when $i = 1$, $\boldsymbol{\epsilon}$ is set as $\mathbf{0}$ to improve the sampling quality.

We mimic the simplified objective proposed by Ho et al. (2020) to train our conditional $\boldsymbol{\epsilon}$-model via

$$\mathcal{L}_d(\theta) = \mathbb{E}_{i \sim \mathcal{U}, \boldsymbol{\epsilon} \sim \mathcal{N}(\mathbf{0}, \boldsymbol{I}), (\boldsymbol{s}, \boldsymbol{a}) \sim \mathcal{D}} \big[ ||\boldsymbol{\epsilon} - \boldsymbol{\epsilon}_\theta(\sqrt{\bar{\alpha}_i}\boldsymbol{a} + \sqrt{1-\bar{\alpha}_i}\boldsymbol{\epsilon}, \boldsymbol{s}, i)||^2 \big], \quad (2)$$

where $\mathcal{U}$ is a uniform distribution over the discrete set as $\{1, \dots, N\}$ and $\mathcal{D}$ denotes the offline dataset, collected by behavior policy $\pi_b$. This diffusion model loss $\mathcal{L}_d(\theta)$ is a behavior-cloning loss, which aims to learn the behavior policy $\pi_b(\boldsymbol{a} \,|\, \boldsymbol{s})$ (i.e. it seeks to sample actions from the same distribution as the training data). Note the marginal of the reverse diffusion chain provides an implicit, expressive distribution that can capture complex distribution properties, such as skewness and multi-modality, exhibited by the offline datasets. In addition, the regularization is sampling-based that only requires taking random samples from both $\mathcal{D}$ and the current policy (i.e. this method does not require us to know the behavior policy, which may be infeasible when the dataset is collected by human demonstrations). Different from the usual two-step strategy, our strategy provides a clean and effective way of applying regularization on a flexible policy.

$\mathcal{L}_d(\theta)$ can efficiently be optimized by sampling a single diffusion step $i$ for each data point, but the reverse sampling in Equation (1), which requires iteratively computing $\boldsymbol{\epsilon}_\theta$ networks $N$ times, can become a bottleneck for the running time. Thus we may want to limit $N$ to a relatively small value. To work with small $N$, with $\beta_{\min} = 0.1$ and $\beta_{\max} = 10.0$, we follow Xiao et al. (2021) to define

$$\beta_i = 1 - \alpha_i = 1 - e^{-\beta_{\min}(\frac{1}{N}) - 0.5(\beta_{\max} - \beta_{\min})\frac{2i-1}{N^2}},$$

which is a noise schedule obtained under the variance preserving SDE of Song et al. (2021).

## 3.2  Q-LEARNING

The policy-regularization loss $\mathcal{L}_d(\theta)$ is a behavior-cloning term, but would not result in learning a policy that can outperform the behavior policy that generated the training data. To improve the policy, we inject Q-value function guidance into the reverse diffusion chain in the training stage in order to learn to preferentially sample actions with high values. The final policy-learning objective is a linear combination of policy regularization and policy improvement:

$$\pi = \arg\min_{\pi_\theta} \mathcal{L}(\theta) = \mathcal{L}_d(\theta) + \mathcal{L}_q(\theta) = \mathcal{L}_d(\theta) - \alpha \cdot \mathbb{E}_{\boldsymbol{s} \sim \mathcal{D}, \boldsymbol{a}^0 \sim \pi_\theta} \big[ Q_\phi(\boldsymbol{s}, \boldsymbol{a}^0) \big]. \quad (3)$$

Note that $\boldsymbol{a}^0$ is reparameterized by Equation (1) and hence the gradient of the Q-value function with respect to the action is backpropagated through the whole diffusion chain.

As the scale of the Q-value function varies in different offline datasets, to normalize it, we follow Fujimoto & Gu (2021) to set $\alpha$ as $\alpha = \frac{\eta}{\mathbb{E}_{(\boldsymbol{s}, \boldsymbol{a}) \sim \mathcal{D}}[|Q_\phi(\boldsymbol{s}, \boldsymbol{a})|]}$, where $\eta$ is a hyperparameter that balances the two loss terms and the Q in the denominator is for normalization only and not differentiated over.

The Q-value function itself is learned in a conventional way, minimizing the Bellman operator (Lillicrap et al., 2015; Fujimoto et al., 2019) with the double Q-learning trick (Hasselt, 2010). We built two Q-networks, $Q_{\phi_1}$, $Q_{\phi_2}$, and target networks $Q_{\phi'_1}$, $Q_{\phi'_2}$ and $\pi_{\theta'}$. We then optimize $\phi_i$ for $i = \{1, 2\}$ by minimizing the objective,

$$\mathbb{E}_{(\boldsymbol{s}_t, \boldsymbol{a}_t, \boldsymbol{s}_{t+1}) \sim \mathcal{D}, \boldsymbol{a}^0_{t+1} \sim \pi_{\theta'}} \left[ \left|\left| \big( r(\boldsymbol{s}_t, \boldsymbol{a}_t) + \gamma \min_{i=1,2} Q_{\phi'_i}(\boldsymbol{s}_{t+1}, \boldsymbol{a}^0_{t+1}) \big) - Q_{\phi_i}(\boldsymbol{s}_t, \boldsymbol{a}_t) \right|\right|^2 \right]. \quad (4)$$

We conduct extensive experiments in Sections 4 and 5 and show that $\mathcal{L}_d$ and $\mathcal{L}_q$ work together to achieve the best performance. We summarize our implementation in Algorithm 1.

---

**Algorithm 1** Diffusion Q-learning

---

Initialize policy network $\pi_\theta$, critic networks $Q_{\phi_1}$ and $Q_{\phi_2}$, and target networks $\pi_{\theta'}$, $Q_{\phi_1'}$ and $Q_{\phi_2'}$
**for** each iteration **do**
    Sample transition mini-batch $\mathcal{B} = \{(s_t, a_t, r_t, s_{t+1})\} \sim \mathcal{D}$.
    *# Q-value function learning*
    Sample $a_{t+1}^0 \sim \pi_{\theta'}(a_{t+1} \,|\, s_{t+1})$ by Equation (1).
    Update $Q_{\phi_1}$ and $Q_{\phi_2}$ by Equation (4). (max Q backup by Kumar et al. (2020) could be added)
    *# Policy learning*
    Sample $a_t^0 \sim \pi_\theta(a_t \,|\, s_t)$ by Equation (1).
    Update policy by minimizing Equation (3).
    *# Update target networks*
    $\theta' = \rho\theta' + (1 - \rho)\theta$, $\phi_i' = \rho\phi_i' + (1 - \rho)\phi_i$ for $i = \{1, 2\}$.
**end for**

---

## 4    POLICY REGULARIZATION

In this section, we illustrate how the previous policy regularization methods work compared to our conditional diffusion-based approach on a simple bandit task with a 2D continuous action space. Below we first provide a brief review of prior methods.

**BC-MLE.** The policy $\pi_\theta(a \,|\, s)$ is modeled by a Gaussian distribution $\mathcal{N}(a; \boldsymbol{\mu}_\theta(s_t), \boldsymbol{\Sigma}_\theta(s))$, where usually $\boldsymbol{\mu}_\theta$ and $\boldsymbol{\Sigma}_\theta$ are parameterized by multi-layer perceptrons (MLPs) and for simplicity $\boldsymbol{\Sigma}_\theta$ is assumed to be a diagonal matrix. The policy is optimized by maximizing $\mathbb{E}_{(s,a)\sim\mathcal{D}}[\log \pi_\theta(a \,|\, s)]$. TD3+BC (Fujimoto & Gu, 2021) directly add the behavior-cloning (BC) loss as an additional term in policy learning, while IQL (Kostrikov et al., 2021b) proposes using "in-sample" learned advantage functions to reweigh the $\log$ term inside the expectation.

**BC-CVAE.** The policy $\pi_\theta(a \,|\, s)$ is modeled by a CVAE model with an encoder network $q_\theta(z \,|\, s, a)$ and a decoder network $p_\theta(a \,|\, s, z)$. The two networks are optimized by maximizing the evidence lower bound $\mathbb{E}_{(s,a)\sim\mathcal{D}}[\mathbb{E}_{z\sim q(\cdot \,|\, s,a)}[\log p(a \,|\, s, z)] - \mathrm{KL}(q(z \,|\, s, a)\|p(z))]$, where $p(z)$ is a prior distribution that is usually set as standard Gaussian. BCQ (Fujimoto et al., 2019) trains a CVAE model as an approximation of the behavior policy and trains another deviation model to guide the actions drawn from the CVAE approximation towards the regions with high learned Q-values.

**BC-MMD.** BEAR (Kumar et al., 2019) also mimics the behavior policy via a CVAE model and proposes to limit the current policy $\pi_\theta(a \,|\, s)$ to be close to the cloned behavior policy via MMD minimization. A Tanh-Gaussian policy is used, which is a Gaussian network with a Tanh activation function at the output layer.

BCQ and BEAR can be seen as a two-step regularization: First, an approximation of the behavior policy is learned (behavior cloning), and then the policy learned by policy improvement is regularized towards the cloned behavior policy. However, such a two-step approach means that the efficacy of the second-step policy regularization heavily depends on the cloning quality, and an inaccurate regularization could misguide the subsequent policy improvement step. We illustrate this weakness in a 2D continuous action space bandit example.

**Example.** We consider a simple bandit task with real-valued actions in a 2D space, $a \in [-1, 1]^2$. We construct an offline dataset $\mathcal{D} = \{(a_j)\}_{j=1}^M$ with $M = 10000$ action examples, where the actions are collected from an equal mixture of four Gaussian distributions with centers $\boldsymbol{\mu} \in \{(0.0, 0.8), (0.8, 0.0), (0.0, -0.8), (-0.8, 0.0)\}$ and standard deviations $\boldsymbol{\sigma}_d = (0.05, 0.05)$, as depicted in the first panel of Figure 1. Note that this example exhibits strong multi-modality in the behavior policy distribution, which is often the case when the dataset is collected by different policies. For example, if multiple humans are involved, some may be experts and choose actions from a different mode from amateur demonstrators.

To evaluate the strength of prior regularization methods, we first compare them to our diffusion-based approach on a behavior-cloning task, where the goal is to just clone the behavior policy that generated the data, not improve on it. As shown in the first row of Figure 1, we observe that the diffusion model captures all the four density modes of the behavior policy. The policy of BC-MLE is limited to a single mode and hence exhibits a strong mode-covering behavior. It fits the four den-

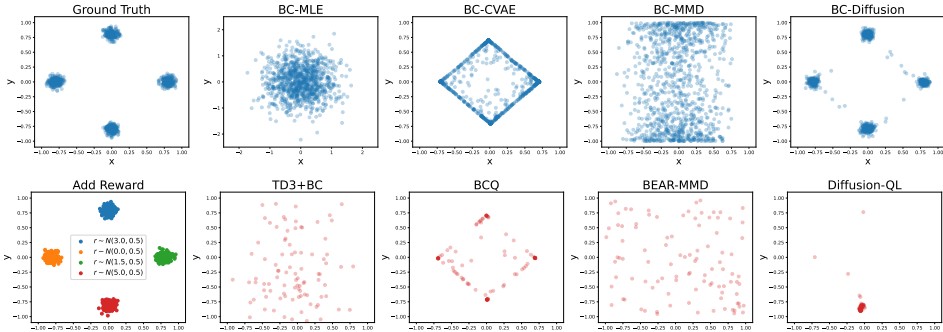

Figure 1: Offline RL experiments on a simple bandit task. The first row shows the comparison of behavior cloning between our method (BC-Diffusion, $N = 50$) and prior methods. Prior methods struggle to capture the multi-modal behavior policy (ground truth). The second row shows the comparison results when policy improvement is added (first figure shows the rewards). The policy regularization of prior methods results in a poor policy, since the behavior-cloning step fails to capture the multi-modal behavior policy, while our method (Diffusion-QL) correctly identifies the high reward behavior mode.

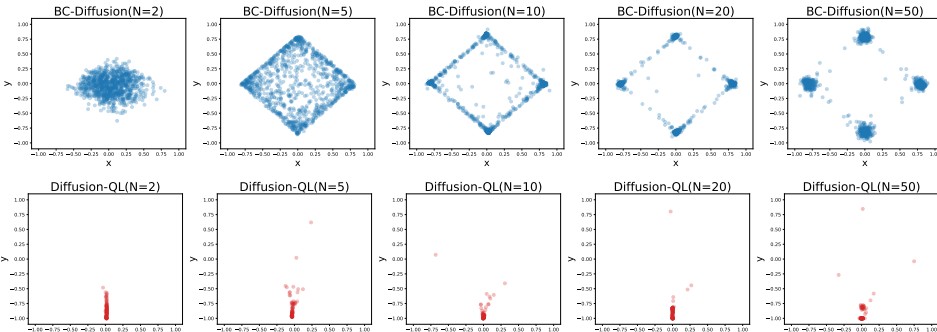

Figure 2: Experiment examining the effect of varying the number of diffusion steps $N$ on the simple bandit talks. The first row shows the ablation study of $N$ for BC-Diffusion. The second row shows the ablation study of $N$ for Diffusion-QL. Large $N$ results in a better fit to the data distribution, but leads to a higher computational cost. By combining the behavior cloning and Q-value losses during training, we are able to learn near-optimal policies with fewer diffusion steps.

sity modes by one Gaussian distribution with a large standard deviation, whose high-density regions are actually the low-density regions of the true behavior policy. The CVAE model is shown to exhibit mode-covering behavior even though it itself is an implicit model with enough expressiveness: We see that CVAE captures the four modes but high densities are assigned between them to cover all the modes. Note we also observe the CVAE model sometimes fails to capture all four modes with different random seeds. The Tanh-Gaussian policy optimized under MMD learns to align the densities near the boundary lines and, due to its limited policy expressiveness, fails to capture the true distribution. These observed failures on behavior cloning in this simple example illustrate the limitations of prior regularization methods in the common case of multi-modal behavior policies.

Next, we investigate how the policy improvement will be impacted by the corresponding policy regularization. We assign each data point a reward sampled from a Gaussian distribution, whose mean is determined by the data center and standard deviation is fixed as 0.5, as shown in the second row of Figure 1. Note here we mimic the offline RL setting, under which the underlying reward function is unknown and needs to be learned. We compare Diffusion Q-learning (QL) with prior methods, including TD3+BC, BCQ, and BEAR-MMD. We train all methods with 1000 epochs to ensure convergence. Due to the strong policy constraint applied by each method, we observe that the policy improvement of prior methods is constrained to suboptimal or even wrong exploration regions, induced by the corresponding behavior-cloning regularization. TD3+BC cannot converge to the optimal mode since the behavior policy places most density in the region where no offline data exists. BCQ learns to place major actions on the four diagonal corners discovered by its CVAE-based behavior cloning. The policy of BEAR-MMD learns to place actions randomly since the exploration region is constrained by inaccurate policy regularization. We observe that the prior regularizations typically push the policy to converge to sub-optimal solutions, such as BCQ, or prevent the policy

from being concentrated on the optimal corner, such as TD3+BC and BEAR-MMD. By contrast, the policy of Diffusion-QL successfully converges to the optimal bottom corner. This is because 1) diffusion policy is expressive enough to recover the behavior policy, which covers all modes for further exploration; 2) the Q-learning guidance, directly through linearly combined loss functions in Equation (3), helps diffusion policy seek optimal actions in the region. The two components are working together to produce good performance.

**Diffusions steps.** We further investigated how the diffusion policy performs as the number of diffusion timesteps $N$ is varied. As expected, the first row of Figure 2 shows that as $N$ increases, the diffusion model becomes more expressive and learns more details about the underlying data distribution. When $N$ is increased to $50$, the true data distribution is accurately recovered. The second row shows that with Q-learning applied, a moderately small $N$ is able to deliver good performance due to our loss coupling defined in Equation (3). However, we can see with a larger $N$ the policy regularization imposed by the diffusion model-based cloning becomes stronger. For example, when $N = 2$, there are still a few action points sampled near regions with no training data, whereas when $N = 50$, the policy is constrained in the correct data region for further exploration. The number of timesteps $N$ serves as a trade-off between policy expressiveness and computational cost for Diffusion-QL. We found $N = 5$ performs well on D4RL (Fu et al., 2020) datasets, which is also a small enough value for cost-effective training and deployment.

## 5 EXPERIMENTS

We evaluate our method on the popular D4RL (Fu et al., 2020) benchmark. Further, we conduct empirical studies on the number of timesteps required for our diffusion model and also perform an ablation study for analyzing the contribution of the two main components of Diffusion-QL.

**Datasets.** We consider four different domains of tasks in D4RL benchmark: Gym, AntMaze, Adroit, and Kitchen. The Gym-MuJoCo locomotion tasks are the most commonly used standard tasks for evaluation and are relatively easy, since they usually include a significant fraction of near-optimal trajectories in the dataset and the reward function is quite smooth. AntMaze consists of more challenging tasks, which have sparse rewards and explicitly need the agent to stitch various sub-optimal trajectories to find a path towards the goal of the maze (Fu et al., 2020). Adroit datasets are mostly collected by human behavior and the state-action region reflected by the offline data is often very narrow, so strong policy regularization is needed to ensure that the agent stays in the expected region. The Kitchen environment requires the agent to complete 4 target subtasks in order to reach a desired state configuration, and hence long-term value optimization is important for it.

**Baselines.** We consider different classes of baselines that perform well in each domain of tasks. For policy regularization-based methods, we include the classic BC, BEAR (Kumar et al., 2019), BRAC (Wu et al., 2019), BCQ (Fujimoto et al., 2019), TD3+BC (Fujimoto & Gu, 2021), AWR (Peng et al., 2019), AWAC (Nair et al., 2020), and IQL (Kostrikov et al., 2021b), along with the Onestep RL (Brandfonbrener et al., 2021), which is based on single-step improvement. For Q-value constraint methods, we include REM (Agarwal et al., 2020) and CQL (Kumar et al., 2020). For model-based offline RL, we consider MoRel (Kidambi et al., 2020). For sequence modelling approaches, we compare with Decision Transformer (DT) (Chen et al., 2021) and Diffuser (Janner et al., 2022). We report the performance of baseline methods either using the best results reported from their own paper, Fu et al. (2020) or Kostrikov et al. (2021b).

**Experimental details.** We train for 1000 epochs (2000 for Gym tasks). Each epoch consists of 1000 gradient steps with batch size 256. The training is usually quite stable as shown in Figure 3 except that we observe the training for AntMaze tasks has variations due to its sparse reward setting and lack of optimal trajectories in the offline datasets. Hence, we save multiple model checkpoints during training and use a completely offline method, as described in Appendix D, to select the best checkpoint for performance evaluation. Using $\mathcal{L}_d$ loss as a lagging indicator of online performance, we perform early stopping and select the checkpoint with the second or third lowest $\mathcal{L}_d$ value. The results in our main paper are all based on the offline model selection. When a small amount of online interaction can be used for model selection, we can achieve even better results (Appendix Table 4).

**Effect of hyperparameter $N$.** We conduct an empirical study on the effect of the number of timesteps $N$ of our diffusion model on real tasks. As shown in Figure 3, empirically we find as

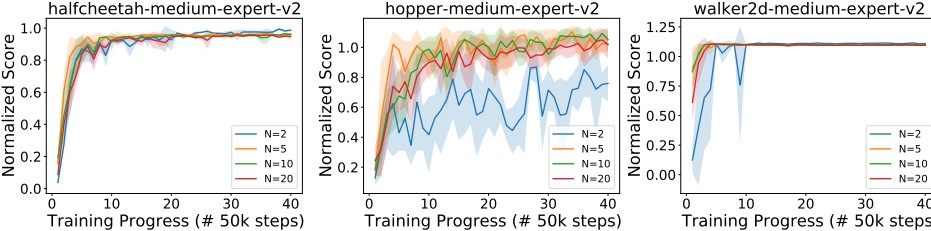

Figure 3: Ablation study of $N$ on selected Gym tasks. We consider a $N$ grid [2, 5, 10, 20] and we find that $N = 5$ is good enough for the selected tasks.

Table 1: The performance of Diffusion-QL and SOTA baselines on D4RL Gym, AntMaze, Adroit, and Kitchen tasks. Results for Diffusion-QL correspond to the mean and standard errors of normalized scores over 50 random rollouts (5 independently trained models and 10 trajectories per model) for Gym tasks, which generally exhibit low variance in performance, and over 500 random rollouts (5 independently trained models and 100 trajectories per model) for the other tasks. Note the standard error of AntMaze is usually large since the return of trajectories is binomial (1 for success while 0 for failure). Our method outperforms all prior methods by a clear margin on all domain, even on the challenging AntMaze, for which behavior cloning methods would fail and some form of policy improvement is essential.

| Gym Tasks | BC | AWAC | Diffuser | MoRel | Onestep RL | TD3+BC | DT | CQL | IQL | Diffusion-QL |
|---|---|---|---|---|---|---|---|---|---|---|
| halfcheetah-medium-v2 | 42.6 | 43.5 | 44.2 | 42.1 | 48.4 | 48.3 | 42.6 | 44.0 | 47.4 | **51.1** ± 0.5 |
| hopper-medium-v2 | 52.9 | 57.0 | 58.5 | **95.4** | 59.6 | 59.3 | 67.6 | 58.5 | 66.3 | 90.5 ± 4.6 |
| walker2d-medium-v2 | 75.3 | 72.4 | 79.7 | 77.8 | 81.8 | 83.7 | 74.0 | 72.5 | 78.3 | **87.0** ± 0.9 |
| halfcheetah-medium-replay-v2 | 36.6 | 40.5 | 42.2 | 40.2 | 38.1 | 44.6 | 36.6 | 45.5 | 44.2 | **47.8** ± 0.3 |
| hopper-medium-replay-v2 | 18.1 | 37.2 | 96.8 | 93.6 | 97.5 | 60.9 | 82.7 | 95.0 | 94.7 | **101.3** ± 0.6 |
| walker2d-medium-replay-v2 | 26.0 | 27.0 | 61.2 | 49.8 | 49.5 | 81.8 | 66.6 | 77.2 | 73.9 | **95.5** ± 1.5 |
| halfcheetah-medium-expert-v2 | 55.2 | 42.8 | 79.8 | 53.3 | 93.4 | 90.7 | 86.8 | 91.6 | 86.7 | **96.8** ± 0.3 |
| hopper-medium-expert-v2 | 52.5 | 55.8 | 107.2 | 108.7 | 103.3 | 98.0 | 107.6 | 105.4 | 91.5 | **111.1** ± 1.3 |
| walker2d-medium-expert-v2 | 107.5 | 74.5 | 108.4 | 95.6 | **113.0** | 110.1 | 108.1 | 108.8 | 109.6 | 110.1 ± 0.3 |
| **Average** | 51.9 | 50.1 | 75.3 | 72.9 | 76.1 | 75.3 | 74.7 | 77.6 | 77.0 | **88.0** |
| **AntMaze Tasks** | BC | AWAC | BCQ | BEAR | Onestep RL | TD3+BC | DT | CQL | IQL | Diffusion-QL |
| antmaze-umaze-v0 | 54.6 | 56.7 | 78.9 | 73.0 | 64.3 | 78.6 | 59.2 | 74.0 | 87.5 | **93.4** ± 3.4 |
| antmaze-umaze-diverse-v0 | 45.6 | 49.3 | 55.0 | 61.0 | 60.7 | 71.4 | 53.0 | **84.0** | 62.2 | 66.2 ± 8.6 |
| antmaze-medium-play-v0 | 0.0 | 0.0 | 0.0 | 0.0 | 0.3 | 10.6 | 0.0 | 61.2 | 71.2 | **76.6** ± 10.8 |
| antmaze-medium-diverse-v0 | 0.0 | 0.7 | 0.0 | 8.0 | 0.0 | 3.0 | 0.0 | 53.7 | 70.0 | **78.6** ± 10.3 |
| antmaze-large-play-v0 | 0.0 | 0.0 | 6.7 | 0.0 | 0.0 | 0.2 | 0.0 | 15.8 | 39.6 | **46.4** ± 8.3 |
| antmaze-large-diverse-v0 | 0.0 | 1.0 | 2.2 | 0.0 | 0.0 | 0.0 | 0.0 | 14.9 | 47.5 | **56.6** ± 7.6 |
| **Average** | 16.7 | 18.0 | 23.8 | 23.7 | 20.9 | 27.3 | 18.7 | 50.6 | 63.0 | **69.6** |
| **Adroit Tasks** | BC | SAC | BCQ | BEAR | BRAC-p | BRAC-v | REM | CQL | IQL | Diffusion-QL |
| pen-human-v1 | 25.8 | 4.3 | 68.9 | -1.0 | 8.1 | 0.6 | 5.4 | 35.2 | 71.5 | **72.8** ± 9.6 |
| pen-cloned-v1 | 38.3 | -0.8 | 44.0 | 26.5 | 1.6 | -2.5 | -1.0 | 27.2 | 37.3 | **57.3** ± 11.9 |
| **Average** | 32.1 | 1.8 | 56.5 | 12.8 | 4.9 | -1.0 | 2.2 | 31.2 | 54.4 | **65.1** |
| **Kitchen Tasks** | BC | SAC | BCQ | BEAR | BRAC-p | BRAC-v | AWR | CQL | IQL | Diffusion-QL |
| kitchen-complete-v0 | 33.8 | 15.0 | 8.1 | 0.0 | 0.0 | 0.0 | 0.0 | 43.8 | 62.5 | **84.0** ± 7.4 |
| kitchen-partial-v0 | 33.8 | 0.0 | 18.9 | 13.1 | 0.0 | 0.0 | 15.4 | 49.8 | 46.3 | **60.5** ± 6.9 |
| kitchen-mixed-v0 | 47.5 | 2.5 | 8.1 | 47.2 | 0.0 | 0.0 | 10.6 | 51.0 | 51.0 | **62.6** ± 5.1 |
| **Average** | 38.4 | 5.8 | 11.7 | 20.1 | 0.0 | 0.0 | 8.7 | 48.2 | 53.3 | **69.0** |

the $N$ increases, the model converges faster and the performance becomes more stable. In the following D4RL tasks, we set a moderate value, $N = 5$, to balance the performance and computational cost. With $N = 5$, the training time of our method is similar to that of CQL (Kumar et al., 2020). The other hyperparameters, such as the learning rate and $\eta$, are provided in Appendix E.

## 5.1 COMPARISON TO OTHER METHODS

We compare our Diffusion-QL with the baselines on four domains of tasks and report the results in Table 1. We give the analysis based on each specific domain.

**Results for Gym Domain.** We can see while most baselines already work well on the Gym tasks, Diffusion-QL can often further improve their performance by a clear margin, especially in 'medium'

Table 2: Ablation study. We conduct an ablation study to compare our diffusion model with CVAE models, and our Q-learning method with BCQ policy improvement.

| Gym Tasks | BC-CVAE | BC-Diffusion | BCQ-CVAE | BCQ-Diffusion | CVAE-QL | Diffusion-QL |
|---|---|---|---|---|---|---|
| halfcheetah-medium-expert-v2 | $67.3 \pm 7.4$ | $76.6 \pm 7.0$ | $96.1 \pm 0.5$ | $94.6 \pm 1.0$ | $70.3 \pm 5.5$ | $\mathbf{97.2 \pm 0.4}$ |
| hopper-medium-expert-v2 | $69.9 \pm 8.6$ | $78.0 \pm 8.9$ | $108.5 \pm 0.6$ | $109.3 \pm 0.9$ | $109.2 \pm 4.6$ | $\mathbf{112.3 \pm 0.8}$ |
| walker2d-medium-expert-v2 | $102.5 \pm 4.4$ | $103.1 \pm 4.4$ | $110.7 \pm 0.2$ | $\mathbf{114.0 \pm 0.5}$ | $50.5 \pm 38.2$ | $111.2 \pm 0.9$ |
| **Average** | 79.9 | 85.9 | 105.1 | 106.0 | 76.6 | **106.9** |

and 'medium-replay' tasks. Note the 'medium' datasets include the trajectories collected by an online SAC (Haarnoja et al., 2018) agent trained to approximately 1/3 the performance of the expert. Hence, the Tanh-Gaussian policy at that time is usually exploratory and not concentrated, which makes the collected data distribution hard to learn. As shown in Section 4, the diffusion model has the expressivity to mimic the behavior policy even in complicated cases and then the policy improvement term will guide the policy to converge to the optimal actions in the subset of explored action space. These two components are key to the good empirical performance of Diffusion-QL.

**Results for AntMaze Domain.** The sparse reward and large amount of sub-optimal trajectories make the AntMaze tasks especially difficult. Strong and stable Q-learning is required to achieve good performance. For example, BC-based methods could easily fail on 'medium' and 'large' tasks. We show that the proposed Q-learning guidance added during the training of a conditional diffusion model is stable and effective. Empirically, with a proper $\eta$, we find Diffusion-QL outperforms the prior methods by a clear margin, especially in harder tasks, such as 'large-diverse'.

**Results for Adroit and Kitchen Domain.** We find the Adroit domain needs strong policy regularization to overcome the extrapolation error (Fujimoto et al., 2019) in offline RL, due to the narrowness of the human demonstrations. With a smaller $\eta$, Diffusion-QL easily beats the other baselines by its reverse diffusion-based policy, which has high expressiveness and better policy regularization. Moreover, long-term value optimization is required for the Kitchen tasks, and we find Diffusion-QL also performs very well in this domain.

## 5.2 ABLATION STUDY

In this section, we analyze why Diffusion-QL outperforms the other policy-constraint based methods quantitatively on D4RL tasks. We conduct an ablation study on the two main components of Diffusion-QL: use of a diffusion model as an expressive policy and Q-learning guidance. For the policy part, we compare our diffusion model with the popular CVAE model for behavior cloning. For the Q-learning component, we compare with the BCQ approach on policy improvement.

As shown in Table 2, BC-Diffusion model outperforms BC-CVAE, validating that our diffusion-based policy is more expressive and better at capturing the data distributions (as we would expect from the results in figure 1). Under the same BCQ framework, which explicitly limits how far the action samples from polices could deviate from the cloned action samples, BCQ-Diffusion still works better than BCQ-CVAE. The hard and physical value constraints on the deviation by BCQ actually limits the policy improvement, as we can see that, Diffusion-QL further boosts the performance. Note Diffusion-QL applies the diffusion model learning itself as a soft policy regularization and guides the policy optimization via additive Q-learning. The weak expressiveness and poor cloning behavior of CVAE makes it fail when coupling with a free Q-learning guidance, as shown by the results of CVAE-QL. In a nutshell, the ablation study shows that the two components of Diffusion-QL are working together to produce good performance.

## 6 CONCLUSION

In this work, we present Diffusion-QL, which is a conditional diffusion-based offline RL algorithm. First, its policy is built by the reverse chain of a conditional diffusion model, which allows for a highly expressive policy class and whose learning itself acts as a strong policy regularization method. Second, Q-learning guidance through a jointly learned Q-value function is injected in the learning of the diffusion policy, which guides the denoising sampling towards the optimal region in its exploration area. The two key components contribute to its state-of-the-art performance across all tasks in the D4RL benchmark.

ACKNOWLEDGEMENTS

Z. Wang and M. Zhou acknowledge the support of NSF-IIS 2212418 and the Texas Advanced Computing Center (TACC) for providing HPC resources that have contributed to the research results reported within this paper.

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

# Appendix

## A  MORE TOY EXPERIMENTS

Here we describe an additional toy experiment on a bandit task. Actions are again in a real-valued 2D space, $a \in [-1, 1]^2$. The offline data $\mathcal{D} = \{(a_i)\}_{i=1}^{10000}$ are collected by sampling actions equally from four Gaussian distributions with centers $\mu \in \{(-0.8, 0.8), (0.8, 0.8), (0.8, -0.8), (-0.8, -0.8)\}$ and standard deviations $\sigma_d = (0.05, 0.05)$, as depicted in the first panel of Figure 4. We conduct the same experiments as the ones in our main paper (see figure 1) and show the performance in Figure 4. The only difference in this experiment is that the samples are now in the corners of the ation space.

For behavior-cloning experiments, we observe that only our diffusion model could recover the original data distribution while the prior regularization methods fail in some way. For example, CVAE could only capture the two diagonal modes and place density between them, while MMD tends to align density around the boundaries because of its Tanh-Gaussian policy. For Q-learning experiments, we observe that the prior regularization methods typically push the policy to converge to sub-optimal solutions in BCQ and BEAR while preventing the policy of TD3+BC from being concentrated on the right corner. However, the policy of Diffusion-QL successfully converges to the optimal bottom corner. The ablation study experiments are consistent with our conclusion in the main paper.

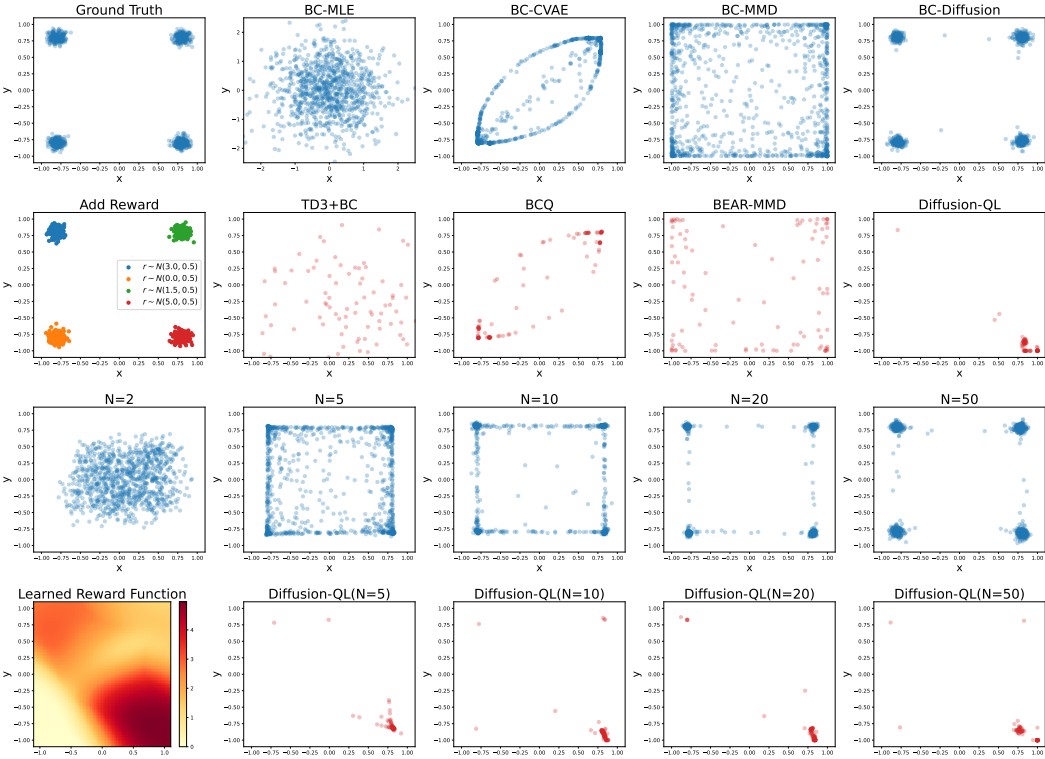

Figure 4: Bandit experiment with a strong multi-modal behavior policy. The first row shows the comparison of behavior-cloning results between our method and prior methods. The second row shows the comparison results with Q-learning involved. The third row shows the ablation study of $N$ for BC-Diffusion. The fourth row shows the learned reward function and the ablation study of $N$ for Diffusion-QL.

## B    IMPLEMENTATION DETAILS

**Diffusion policy.** We build our policy as an MLP-based conditional diffusion model. Following the parameterization of Ho et al. (2020), the model itself is a residual model, $\epsilon_\theta(\boldsymbol{a}^i, \boldsymbol{s}, i)$, where the $i$ is the last timestep and $\boldsymbol{s}$ is the state condition. We model $\epsilon_\theta$ as a 3-layer MLPs with Mish activations and we use 256 hidden units for all networks. The input of $\epsilon_\theta$ is the concatenation of the last step action vector, the current state vector, and the sinusoidal positional embedding of timestep $i$. The output of $\epsilon_\theta$ is the predicted residual at diffusion timestep $i$.

**Q networks.** We build two Q networks with the same MLP setting as our diffusion policy, which has 3-layer MLPs with Mish activations and 256 hidden units for all networks.

We use the Adam (Kingma & Ba, 2014) optimizer for the training of both Diffusion policy and Q networks.

## C    EXPERIMENTAL DETAILS

We train our algorithm with 2000 epochs for Gym tasks and 1000 epochs for the other tasks, where each epoch consists of 1000 gradient steps. For the Gym locomotion tasks, we average mean returns over 6 independently trained models and 10 trajectories per mode. For the other tasks, we average over 6 independently trained models and 100 evaluation trajectories.

We investigate two model selection methods, online and offline model selection. For online model selection, following the convention of supervised learning, the best-performed model is saved during training and used for final evaluation. For offline case, we select the model based on our lagging indicator $\mathcal{L}_d$ and more details are in Appendix D.

We use the original task rewards from MuJoCo Gym and Kitchen tasks. We standardize Adroit task rewards for training stability. We modify the rewards according to the suggestion of CQL (Kumar et al., 2020) for the AntMaze datasets.

## D    OFFLINE MODEL SELECTION

For reducing the training cost and picking the best model during training without any interaction with the real environment, we provide a way to properly conduct early stopping for Diffusion-QL. Empirically, we found that $\mathcal{L}_d$ loss is a lagging indicator of online performance. Note $\mathcal{L}_d$ is the behavior cloning loss of our diffusion model-based policy and measures how close the current policy is to the behavior policy. For offline RL, we want $\mathcal{L}_d$ to be small to get rid of the OOD issue but we also don't expect $\mathcal{L}_d$ to be the smallest since our goal is policy learning while not behavior cloning. Hence, we monitor the $\mathcal{L}_d$ and save multiple model checkpoints during training. We stop the training whenever $\mathcal{L}_d$ increases in our evaluation stage for early stopping. When the training stopped, we picked the checkpoint according to our metric, the second or third lowest $\mathcal{L}_d$ value. Note our model selection part is totally offline and only based on $\mathcal{L}_d$ without any access to the environment. The selection of the $\mathcal{L}_d$ is not very sensitive. Always using the 2nd smallest checkpoints doesn't impact the performance much. For example, with 2nd checkpoints selected, for the Gym domain, the average score is 87.6, and for the AntMaze domain, the average score is 69.1.

## E    HYPERPARAMETERS

For Diffusion-QL, we consider three hyperparameters in total: learning rate, Q-learning weight $\eta$, and whether to use max Q backup from CQL (Kumar et al., 2020). For the learning rate of Adam, we consider values in the grid $\{1 \times 10^{-3}, 3 \times 10^{-4}, 3 \times 10^{-5}\}$ for the policy, while we use a fixed learning rate, $3 \times 10^{-4}$, for Q-networks. For $\eta$, we consider values according to the characteristics of different domains, as we mentioned in the description of datasets that the Adroit and Kitchen tasks require more policy regularization and the AntMaze tasks require more Q-learning. For max Q backup, we only apply it on the AntMaze tasks. Based on these considerations, we provide our hyperparameter setting in Table 3.

Table 3: Hyperparameter settings of all selected tasks.

| Tasks | learning rate | $\eta$ | max Q backup |
|---|---|---|---|
| halfcheetah-medium-v2 | $3 \times 10^{-4}$ | 1.0 | False |
| hopper-medium-v2 | $3 \times 10^{-4}$ | 1.0 | False |
| walker2d-medium-v2 | $3 \times 10^{-4}$ | 1.0 | False |
| halfcheetah-medium-replay-v2 | $3 \times 10^{-4}$ | 1.0 | False |
| hopper-medium-replay-v2 | $3 \times 10^{-4}$ | 1.0 | False |
| walker2d-medium-replay-v2 | $3 \times 10^{-4}$ | 1.0 | False |
| halfcheetah-medium-expert-v2 | $3 \times 10^{-4}$ | 1.0 | False |
| hopper-medium-expert-v2 | $3 \times 10^{-4}$ | 1.0 | False |
| walker2d-medium-expert-v2 | $3 \times 10^{-4}$ | 1.0 | False |
| antmaze-umaze-v0 | $3 \times 10^{-4}$ | 0.5 | False |
| antmaze-umaze-diverse-v0 | $3 \times 10^{-4}$ | 2.0 | True |
| antmaze-medium-play-v0 | $1 \times 10^{-3}$ | 2.0 | True |
| antmaze-medium-diverse-v0 | $3 \times 10^{-4}$ | 3.0 | True |
| antmaze-large-play-v0 | $3 \times 10^{-4}$ | 4.5 | True |
| antmaze-large-diverse-v0 | $3 \times 10^{-4}$ | 3.5 | True |
| pen-human-v1 | $3 \times 10^{-5}$ | 0.15 | False |
| pen-cloned-v1 | $3 \times 10^{-5}$ | 0.1 | False |
| kitchen-complete-v0 | $3 \times 10^{-4}$ | 0.005 | False |
| kitchen-partial-v0 | $3 \times 10^{-4}$ | 0.005 | False |
| kitchen-mixed-v0 | $3 \times 10^{-4}$ | 0.005 | False |

## F  OPTIMAL RESULTS

If a small amount of online experience is provided during the evaluation stage for model selection, we can pick the best models during training via online evaluations (similar to early stopping in supervised learning). This regime provides a further boost in the performance of Diffusion-QL as shown in Table 4.

## G  LIMITATIONS AND FUTURE WORK

Diffusion policies are highly expressive and hence they can capture multi-modal distributions well. We have shown this results in learning better policies in offlineRL. However, at the inference time, the reverse sampling defined in Equation (1) requires iteratively computing $\epsilon_\theta$ networks $N$ times, and this can become a bottleneck for the running time. In our case, we couple the learning of diffusion policies with Q-learning, and achieve good performance with small $N = 5$. Diffusion policies with $N = 5$ could be four to five times slower in action inference compared to previous one-step feedforward policies, and hence the inference cost could prevent the approach from deployment in some real-world scenarios, where fast response is necessary.

This motivates potential future works. There have been many recent works focusing on improving the sampling speed of diffusion models (Lu et al., 2022; Wang et al., 2022; Xiao et al., 2021; Salimans & Ho, 2022; Song et al., 2020; Zheng et al., 2022), which could be applied to improve the sampling efficiency of Diffusion-QL. For example, the diffusion policy may be able to be distilled into a simpler feedforward policy after training.

## H  GAUSSIAN MIXTURE POLICY

We test a Gaussian Mixture policy as a classic policy class that can capture multi-modal distributions as an additional baseline. We modify TD3+BC (Fujimoto & Gu, 2021) by replacing the original deterministic actor with a mixture density network (Bishop, 1994), where each mixture component is a Gaussian. Since a Gaussian mixture policy is applied, we replaced minimizing the L2 loss (from TD3+BC) between predicted actions and real actions, with maximizing the likelihood estimate of Gaussian mixtures on real state-action pairs. We keep all the other parts the same as TD3+BC.

Table 4: Performance comparison with online model selection and offline model selection.

| Gym Tasks | Diffusion-QL (Offline) | Diffusion-QL (Online) |
|---|---|---|
| halfcheetah-medium-v2 | $51.1 \pm 0.5$ | $\mathbf{51.5} \pm 0.3$ |
| hopper-medium-v2 | $90.5 \pm 4.6$ | $\mathbf{96.6} \pm 3.4$ |
| walker2d-medium-v2 | $87.0 \pm 0.9$ | $\mathbf{87.3} \pm 0.5$ |
| halfcheetah-medium-replay-v2 | $47.8 \pm 0.3$ | $\mathbf{48.3} \pm 0.2$ |
| hopper-medium-replay-v2 | $101.3 \pm 0.6$ | $\mathbf{102.0} \pm 0.4$ |
| walker2d-medium-replay-v2 | $95.5 \pm 1.5$ | $\mathbf{98.0} \pm 0.5$ |
| halfcheetah-medium-expert-v2 | $96.8 \pm 0.3$ | $\mathbf{97.2} \pm 0.4$ |
| hopper-medium-expert-v2 | $111.1 \pm 1.3$ | $\mathbf{112.3} \pm 0.8$ |
| walker2d-medium-expert-v2 | $110.1 \pm 0.3$ | $\mathbf{111.2} \pm 0.9$ |
| **Average** | 88.0 | **89.3** |
| **AntMaze Tasks** | **Diffusion-QL (Offline)** | **Diffusion-QL (Online)** |
| antmaze-umaze-v0 | $93.4 \pm 3.4$ | $\mathbf{96.0} \pm 3.3$ |
| antmaze-umaze-diverse-v0 | $66.2 \pm 8.6$ | $\mathbf{84.0} \pm 10.1$ |
| antmaze-medium-play-v0 | $76.6 \pm 10.8$ | $\mathbf{79.8} \pm 8.7$ |
| antmaze-medium-diverse-v0 | $78.6 \pm 10.3$ | $\mathbf{82.0} \pm 9.5$ |
| antmaze-large-play-v0 | $46.4 \pm 8.3$ | $\mathbf{49.0} \pm 9.4$ |
| antmaze-large-diverse-v0 | $56.6 \pm 7.6$ | $\mathbf{61.7} \pm 8.2$ |
| **Average** | 69.6 | **75.4** |
| **Adroit Tasks** | **Diffusion-QL (Offline)** | **Diffusion-QL (Online)** |
| pen-human-v1 | $72.8 \pm 9.6$ | $\mathbf{75.7} \pm 9.0$ |
| pen-cloned-v1 | $57.3 \pm 11.9$ | $\mathbf{60.8} \pm 11.8$ |
| **Average** | 65.1 | **68.3** |
| **Kitchen Tasks** | **Diffusion-QL (Offline)** | **Diffusion-QL (Online)** |
| kitchen-complete-v0 | $84.0 \pm 7.4$ | $\mathbf{84.5} \pm 6.1$ |
| kitchen-partial-v0 | $60.5 \pm 6.9$ | $\mathbf{63.7} \pm 5.2$ |
| kitchen-mixed-v0 | $62.6 \pm 5.1$ | $\mathbf{66.6} \pm 3.3$ |
| **Average** | 69.0 | **71.6** |

Table 5: Performance comparison for Gaussian Mixture ablation study.

| Gym Tasks | TD3+BC | TD3+BC-GM | Diffusion-QL |
|---|---|---|---|
| halfcheetah-medium-expert-v2 | 90.4 | 49.0 | **96.8** |
| hopper-medium-expert-v2 | 98.0 | 40.0 | **111.1** |
| walker2d-medium-expert-v2 | 110.1 | 66.5 | **110.1** |

We evaluated the model (we call it TD3+BC-GM) on both our bandit toy experiment and selected D4RL tasks. As shown in Figure 5, with a properly selected number of mixtures, the Gaussian mixture could capture the multi-modal distribution in our behavior cloning experiment. However, in the Q-learning experiment, it fails to converge to the optimal target location, and always places some density on the suboptimal modes, resulting in a suboptimal policy with multi-modes. For D4RL experiments, we set the number of mixtures to be 3, and seen in Table 5, we observed that TD3+BC-GM does not perform well on the three D4RL tasks, which is consistent with our previous observation that TD3+BC-GM is prone to have suboptimal actions.

One reason for the poor performance could be that Gaussian mixture models are challenging to fit well, particularly in higher dimensional spaces. We are also forced to choose the number of mixture components, which for the D4RL experiments we don't know a priori how many components there are. Moreover, each mixture component is often parameterized with a diagonal Gaussian that has limited ability in capturing the dependence between different dimensions.

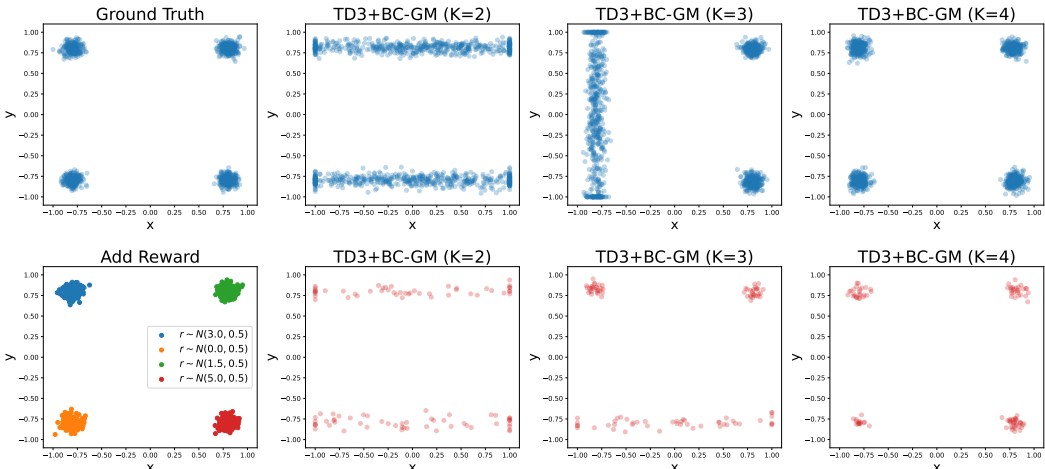

Figure 5: Gaussian Mixture ablation study. The first row shows the behavior cloning experiment and the second row shows the Q-learning experiment, which are the same set of experiments described in Section 4. Here, $K$ is the number of Gaussian mixtures.

