# OpenReview forum: "Diffusion Policies as an Expressive Policy Class for Offline Reinforcement Learning"
_ICLR.cc/2023/Conference — ICLR 2023 poster_

### Official Review · Reviewer_Qge1 · 2022-10-25

**Confidence:** 4
**Correctness:** 2
**Technical Novelty And Significance:** 2
**Empirical Novelty And Significance:** 3
**Recommendation:** 6

**Clarity, Quality, Novelty And Reproducibility:**

Clarity is good, but novelty is limited and reproducibility is questionable. (See weaknesses above)

**Strength And Weaknesses:**

Strengths
- Writing is clear
- Good empirical performance

Weakness
- Limited novelty: the better performance mainly comes from a different parametrization of the policy borrowed from the generative model literature. It is a straightforward application.
- Some of the comparisons in the synthetic experiment are questionable. It is surprising to see that CVAE and MMD methods compared in Sec.5 fail so miserably for such a simple task. It is very likely that these methods are not tuned properly for the bandit problem (e.g., it is not clear why CVAE produces a diamond shape policy) No enough implementation detail is provided here so it is challenging to see what may have been done incorrectly.
- Some of the experiment details are missing and require further clarification. (a) The trade-off hyper-parameter is chosen on a dataset-to-dataset basis, which makes the applicability questionable. It is mentioned in Sec.6.1 and the appendix that eta (also see the discussion after Eq.(3)) is chosen differently for different benchmark problems. The actual values of eta are also not provided in the paper. (b) Page 7 mentioned that L_d is a “lagging indicator” and 2nd or 3rd checkpoints are chosen. This is not convincing at all and additional explanation is needed. As a result, the better performance becomes less convincing since it can be due to over-tuning.

**Summary Of The Paper:**

This paper proposes to use a (conditional) diffusion model to represent a policy for the RL agent in the offline RL setting. The objective (3) consists of a Q-learning term and a behavior regularization term. Synthetic experiments on a bandit problem show the advantages of using diffusion models over other generative models, and experiments on common offline RL benchmarks show that the proposed diffusion-QL method can outperform alternative methods when tuned right.

**Summary Of The Review:**

The paper proposes to use a diffusion model to parametrize the policy and claim that it can achieve better performance in the offline RL setting. However, the application is straightforward. It does not really show why such models are suitable for the task (the synthetic experiments are questionable) and important experiment details are missing. Hence the score.

---

> ### Author Response · Authors · 2022-11-10
> **Response Continued**
>
> > Some of the experiment details are missing and require further clarification. (a) The trade-off hyper-parameter is chosen on a dataset-to-dataset basis, which makes the applicability questionable. It is mentioned in Sec.6.1 and the appendix that eta (also see the discussion after Eq.(3)) is chosen differently for different benchmark problems. The actual values of eta are also not provided in the paper.
>
> The values of $\eta$, as used for experiments, are provided in Table 3. We think the tuning for the trade-off hyperparameter is reasonable, since the scale and sparsity of the rewards varies across tasks. As shown in Table 3, we generally keep the $\eta$ to be the same value for tasks in one domain, such as Gym, Adroit and Kitchen. The AntMaze is an exception because different tasks in the AntMaze have significant differences. The maze size and data distribution changes a lot in different tasks.
>
> We note that prior works also tune hyperparameters on each dataset respectively, for example, CQL selects Lagrange threshold $\tau$ (this hyperparameter also trades off matching the behavior policy vs returns) from a list of values for different tasks, COMBO conducts hyperparameter tuning for every Gym tasks in its Table 4 (where we keep the same across all tasks), and MoRel also conducts hyperparameter tuning for every Gym tasks based on its Table 4 and 5.
>
> > Some of the experiment details are missing and require further clarification. (b) Page 7 mentioned that L_d is a “lagging indicator” and 2nd or 3rd checkpoints are chosen. This is not convincing at all and additional explanation is needed. As a result, the better performance becomes less convincing since it can be due to over-tuning.
>
> $L_d$ as a lagging indicator is mainly for early stopping to reduce the training cost. Always using the 2nd checkpoints doesn’t impact the performance much. For example, with 2nd checkpoints selected, for the Gym domain, the average score is 87.6, and for the AntMaze domain, the average score is 69.1.
>
> > reproducibility is questionable.
>
> We provide all the source code for our experiments including all the hyperparameters used. We have done our best to include relevant details in the appendix. As another reviewer pointed out, we didn’t include the details of the optimizer we used (Adam) in the paper, which we have now added to the appendix. If there is a specific piece of information that you think would help others reproduce this work, please let us know during the discussion period and we will do our best to make sure it is included. For the simple bandit tasks we provide code for all baselines, for the D4RL we use other’s reported performance on these tasks and provide the full code to reproduce our method.

---

> > ### Comment · Reviewer_Qge1 · 2022-11-19
> > **Over-tuning Hyperparameters**
> >
> > I thank the authors for the additional explanations. I agree that it could be beneficial to use a diffusion model to parametrize a policy due to better expressivity. It may also inspire some other researchers to explore this direction further. Therefore, I increased my score. However, I want to emphasize that the empirical results could be misleading. After reading some of the codes, it becomes even more obvious that the hyper-parameters could be over-tuned. In addition to eta and the cherry-picked checkpoints, the gradient norm is also specific to each dataset, which is not discussed in the paper. This creates a misleading impression that diffusion models are almost always superior, but in reality, it can be due to over-tuning.
> >
> > As a minor point, for the toy example, I suspect that the diamond shape of CVAE comes from the clipping of the base distribution rather than due to "strong mode covering".

---

> > > ### Author Response · Authors · 2022-11-19
> > > **Thank you and additional clarifications**
> > >
> > > We appreciate Reviewer Qge1 for acknowledging the novelty and contribution of our work and increasing the score. Back to your concerns on hyperparameter tuning, we add more clarifications below. First, as we answered above, prior offline RL algorithms all rely on more or less hyperparameter tuning to achieve good empirical performance across tasks, and there is no exception to this practice for Diffusion-QL. Second, the hyperparameter tuning for Diffusion-QL mainly is designed for AntMaze tasks due to their sparse reward settings and lack of optimal trajectories in the datasets. For Gym (including 9 tasks), we could easily fix the same set of hyperparameters across all tasks, such as $\eta=1.0$ without grad norm and with early stopping at second lowest $L_d$ (i.e., our training stops when the $L_d$ starts to rise; we select the second lowest among the checkpoints saved before stopping). If the training cost is not a concern, the early stopping could also be ignored here. Grad norm is a general trick for stabilizing the training, which has been widely used for RL algorithms, such as [1] and [2]. We agree that diffusion models are not always superior, and adding extra objectives in diffusion models, such as our Q-learning guidance, could benefit from more sophisticated designs.
> > >
> > > For your minor comments, we thank you for providing potential reasons for explaining the diamond shape of CVAE inherited from BCQ. We will investigate more here and try to make it more clear in our next revision.
> > >
> > > [1] Bjorck, Nils, Carla P. Gomes, and Kilian Q. Weinberger. "Towards Deeper Deep Reinforcement Learning with Spectral Normalization." Advances in Neural Information Processing Systems 34 (2021): 8242-8255.
> > >
> > > [2] Bjorck, Johan, Carla P. Gomes, and Kilian Q. Weinberger. "Is High Variance Unavoidable in RL? A Case Study in Continuous Control." arXiv preprint arXiv:2110.11222 (2021).

---

> ### Author Response · Authors · 2022-11-10
> **Response**
>
> We thank reviewer Qge1 for the comments and suggestions. Below, we address the concerns raised in your review point by point. Please let us know if you have any further concerns or whether this adequately addresses all the issues that you raised with the paper.
>
> > Limited novelty: the better performance mainly comes from a different parametrization of the policy borrowed from the generative model literature. It is a straightforward application.
>
> Applying diffusion models onto offline RL is novel. It is not the case that it is a trivial change (in fact, we considered a few approaches before landing on the method described in the paper). The diffusion model we built is not the same as DDPM. We tried different network architectures, and multiple noise schedules to make it work in offline RL. The coupling of Q-learning guidance onto the final action output of diffusion models is also novel and important for allowing us to use only a small number of diffusion steps for efficient training and inference.
>
> We also kindly point to the novelty discussion from the other reviewers, and hope they can also help address your concern here.
>
> > Some of the comparisons in the synthetic experiment are questionable. It is surprising to see that CVAE and MMD methods compared in Sec.5 fail so miserably for such a simple task. It is very likely that these methods are not tuned properly for the bandit problem (e.g., it is not clear why CVAE produces a diamond shape policy) No enough implementation detail is provided here so it is challenging to see what may have been done incorrectly.
>
> We think there is a misunderstanding here. Since our target is to compare policy optimization with reward provided, we borrowed the CAVE and MMD parts from the selected baseline methods, such as BCQ and BEAR. For example, as shown in Fig 1, the CVAE training is inherited from the BCQ and the MMD training is borrowed from BEAR. To make a fair comparison, we also make each layer of neural networks to have the same number of hidden units, 128.
>
> What we are going to show here is the exact policy regularization that is applied in the prior methods, and we found them inaccurate. The CVAE might not be in the best config for behavior cloning but it was tuned by the original authors for policy optimization. CVAE produces a diamond shape policy due to its strong mode covering behavior.
>
> The empirical evaluation on D4RL benchmarks also coincides with our bandit experiments, in which BCQ and BEAR perform poorly on some datasets. We also include the ablation study in Section 6.2 to illustrate the effectiveness of diffusion models as expressive policies. We directly replace the diffusion models in our algorithm with CVAE and the performance drops significantly on halfcheetah/walker2d-medium-expert-v2.
>
> We provide the code used in these experiments and we will add more detail in the appendix of the hyperparameters used (these are already in the code but we will endeavor to make them clearer).

---

### Official Review · Reviewer_JRTd · 2022-10-25

**Confidence:** 4
**Correctness:** 4
**Technical Novelty And Significance:** 3
**Empirical Novelty And Significance:** 3
**Recommendation:** 8

**Clarity, Quality, Novelty And Reproducibility:**

* Well written
* Good execution
* Novel enough
* Good contribution

**Strength And Weaknesses:**

The paper is well-written and easy to follow. The proposed model is (very) simple -- which is a good point in my mind -- and I am sure that this paper is bringing interesting information to the community. If the idea is almost obvious (using the best-known and trendy generative model instead of simple gaussian models), I think that such a contribution provides important insights and researchers will be able to build on it. What the paper suggests is that the way we are used to capturing the distribution of actions (gaussian conditioned on the state) is not the right one since it is not able to capture multiple modes (while datasets, and particularly medium ones in D4RL are multi-modal since they have been built by using different acquisition policies). On that point, it seems particularly true for BC approaches where two experts may have taken two different actions in a similar state (and the average action would not be an expert action). What is less clear to me is that this also holds when the reward is available (particularly dense reward) where we can expect one action to 'dominate' all the other actions. For instance, in the toy problem (Figure 1), I don't understand why TD3+BC is not able to capture the same distribution as the one captured by Diffusion-QL, even using a simple gaussian. I would be interested in having more explanations on this point.

One question I also have is what would happen when using classical state-of-the-art methods with a mixture of Gaussians instead of a simple gaussian. Indeed, all the comparisons are made with a simple one-mode gaussian, and reimplementing a BC for instance with a mixture of 2 or 3 gaussians would give insight into the need to use diffusion models while simple approaches could work better. I would be very happy to see this kind of experiment in the ablation study for instance, but also in the toy environment which seems to be solvable through a mixture of Gaussians. Author could also compare their approach to 'implicit behavioral cloning' which proposes a solution to the problem of capturing multiple modes in BC.

Minor remark: figures are too small, and axis are not readable.

**Summary Of The Paper:**

The paper proposes a new offline RL algorithm where the distribution of actions (conditionally to the state) is represented by a diffusion model. At the same time, existing methods usually use a simple gaussian. The approach is then quite simple: the diffusion can be used with a classical behavioral cloning loss (diffusion BC), and coupled with a Q function when reward is available (Diffusion QL). When associated with a Q function, the authors use a strategy that is similar to the TD3+BC method proposed in previous work. The method is then evaluated both on a toy bandit environment to better understand the differences between this model and the existing ones, and also on standard offline-RL benchmarks (D4RL). The diffusion algorithm is obtaining very good results and is very competitive w.r.t state-of-the-art.

**Summary Of The Review:**

This is a paper that is solid on both the idea and the execution of the idea. I do not see clear reasons to refuse such a paper.

---

> ### Author Response · Authors · 2022-11-10
> **Response**
>
> We thank reviewer JRTd for providing a positive feedback and helpful suggestions. Below please find a response.
>
> > What is less clear to me is that this also holds when the reward is available (particularly dense reward) where we can expect one action to 'dominate' all the other actions. For instance, in the toy problem (Figure 1), I don't understand why TD3+BC is not able to capture the same distribution as the one captured by Diffusion-QL, even using a simple gaussian. I would be interested in having more explanations on this point.
>
> The intuition we have here is that during training the model needs to first capture the multiple modes in to properly regularize the policy, even when the final optimal policy may be unimodal. TD3+BC while it could, in principle, model the optimal policy, exhibits mode-covering behavior early on in training that prevents it converging to this solution.
>
> > One question I also have is what would happen when using classical state-of-the-art methods with a mixture of Gaussians instead of a simple gaussian.
>
> Thanks for suggesting this. Diffusion models are more expressive than mixture density networks (Bishop, C.M., 1994)
>
> Gaussian mixture networks can be difficult to fit, for example, the number of components has to be chosen as a hyperparameter,  is hard to choose, the covariance matrix can be difficult to learn, and they tend to perform poorly in high dimensional action space.
>
> Following your suggestion, we tried replacing the deterministic actor in TD3+BC with a mixture density network, where each mixture component is a diagonal Gaussian. Since a Gaussian mixture policy is applied, we replaced minimizing the L2 loss (from TD3+BC) between predicted actions and real actions, with maximizing the likelihood estimate of Gaussian mixtures on real state-action pairs. We keep all the other parts the same as TD3+BC.
>
> We evaluated the model (we call it TD3+BC-GM) on both our bandit toy experiment and D4RL tasks. We added Appendix G for more details of the empirical study. Below, we give a brief summary.
>
> For results of the toy experiment, please see Fig. 5 in Appendix G. We found that with a properly selected number of mixtures, the Gaussian mixture could capture the multi-modal distribution in our behavior cloning experiment. However, it did not always do so reliably and when we added policy improvement it fails to converge to the optimal target location, and always places some density on the suboptimal modes.
>
> We also provide the experiment results on D4RL below. We observed that TD3+BC-GM doesn’t perform well on D4RL tasks, which could be due to the convergence issue found in toy experiments and also the high dimensional action space. Note even though a mixture of diagonal Gaussian could in theory capture multi-modes, it has limited ability in capturing the dependence between different action dimensions.
>
> Note that we do also compare with other methods that, in principle, should be able to model multi-modal distributions (BCQ and BEAR both of which use a conditional variational autoencoder for modeling the distribution) but although these are better able to capture multi-modes they do poor job of modeling the distributions.
>
> | Env                          | TD3+BC | TD3+BC-GM | Diffusion-QL |
> |------------------------------|--------|-----------|--------------|
> | halfcheetah-medium-expert-v2 | 90.4   | 48.97     | **96.8**     |
> | hopper-medium-expert-v2      | 98.0   | 40.01     | **111.1**    |
> | walker2d-medium-expert-v2    | 110.1  | 66.50     | **110.1**    |
>
>
> Bishop, C.M., 1994. Mixture density networks.

---

> > ### Comment · Reviewer_JRTd · 2022-11-15
> > **Response**
> >
> > Thanks for the answers to my questions. I would tend to think that the mixture of gaussians need more investigations,  but I appreciate the preliminary experiments and findings.
> >
> > I still consider that this paper is interesting for the community, and support its acceptance.

---

### Official Review · Reviewer_J2jU · 2022-10-26

**Confidence:** 4
**Correctness:** 4
**Technical Novelty And Significance:** 3
**Empirical Novelty And Significance:** 2
**Recommendation:** 8

**Clarity, Quality, Novelty And Reproducibility:**

**Clarity.** The paper is very easy to read, well organised, self-contained and with clear explanations.

**Quality.** The toy example supports the main hypothesis of the paper: diffusion models are able to model multimodal policies, which allows the Q-function to discover the in-sample highest value actions. The results in the D4RL benchmark are convincing that this is a promising approach.

**Novelty.** On one side, the paper just combines diffusion models with a standard critic, and this is motivated from the realisation that the data distribution in offline RL can be rich and multimodal, which has already been discussed in previous works. On the other hand, the paper combines these two ideas in a simple, elegant, and effective manner. The result is a novel model for offline RL and will likely inspire further research, for example using other architectures beyond MLPs, condition on state-action sequences for tackling partial observability, replacing the Q-function with advantage estimates to guide the gradient, other noisy schemes, reducing the sampling cost, etc.

**Reproducibility.** The authors describe their architecture and hyperparameters in detail - I think they missed to give details about the optimizer though. But they have provided their code, which I expect (I haven't actually tried) will make the experiments easily and fully reproducible.

**Strength And Weaknesses:**

 ### Strength
- The paper presents a fresh idea for offline-RL that fixes multiple issues from previous approaches.
- Identifying the main problem with previous policy regularisation approaches is the inaccurate approximation of the data distribution is insightful and leveraging diffusion models for directly modelling multimodal behaviour policies is an effective and sensible approach.
- Experimental results on the D4RL benchmark are consistent and very promising.
- The paper is well written and easy to read.

### Weaknesses
- Some important baselines are missing in the experiment sections that were designed to deal with multimodal data distributions, like the Behaviour Transformers (Shafiullah et al., 2022).
- The paper barely discusses the limitations of the proposed approach, like the high computational cost of sampling actions, which might prevent the approach from deployment in some real-world scenarios.


**Summary Of The Paper:**

The paper presents Diffusion-QL, an offline RL method that can be seen as an actor-critic method, where the actor is a generative diffusion models, and the critic is a standard Q-function. The diffusion model is trained with supervised learning to imitate the data distribution. The main benefit of diffusion models is that they are able to accurately model the data distribution, even when it is multimodal, which is usually a challenge for previous methods. The Q-function is then used to guide the gradient during the training of the diffusion model towards actions with higher Q value. This combined approach prevents the agent from taking out-of-sample actions, while ensuring convergence towards the optimal actions among the explored regions of the state-action set. The authors introduce their approach as a policy regularisation method as it is constrained to a subset of the support of the data distribution.

First, the method is illustrated on a toy bandit problem showing how it is the only one able to accurately model the data distribution and the only one to reliably converge towards the optimal action among the baselines. Finally, further experiments show Diffusion-QL achieves competitive or state of the art results on the D4RL benchmark.

**Summary Of The Review:**

This is an interesting paper that tackles an important and difficult problem, offline-RL, in a novel, elegant and effective manner. I believe the paper presents a proof of concept that can be extended in several ways and will likely inspire further research.

---

> ### Author Response · Authors · 2022-11-10
> **Response**
>
> We thank reviewer J2jU for providing a positive feedback and noting the contributions of our work. Below please find more clarifications.
>
> > Some important baselines are missing in the experiment sections that were designed to deal with multimodal data distributions, like the Behaviour Transformers (Shafiullah et al., 2022).
>
> Thanks for pointing out this related work. Note Shafiullah et al. (2022) is a very recent paper, which was first put online on June 22, 2022, so we missed it during our literature review. The Behavior Transformers (Shafiullah et al., 2022) paper is focused on behavior cloning. It is evaluated on behavior cloning datasets and compares only with other behavior cloning baselines, and hence is not directly comparable over the Offline RL benchmarks used in our work.
>
> Behavior cloning has limited performance in Offline RL benchmarks due to the lack of policy optimization (Fu, J., Kumar et al., 2020), therefore we would not expect it to perform well on tasks that require policy improvement (the majority of tasks we tested on). We did compare with the Decision Transformer (Chen, L et al., 2021). Decision Transformer also proposes to use a transformer-based sequence model as the policy and include a policy optimization part by return guidance. We consider Decision Transformer as a good representative of this class of offline RL algorithms.
>
> However, the Behavior Transformers work also highlights the importance of modeling multi-modal distributions well when learning from demonstrations, we have now cited it when we discuss this point in our paper and thank the reviewer for bringing this work to our attention.
>
> Fu, J., Kumar, A., Nachum, O., Tucker, G. and Levine, S., 2020. D4rl: Datasets for deep data-driven reinforcement learning. arXiv preprint arXiv:2004.07219.
>
> Chen, L., Lu, K., Rajeswaran, A., Lee, K., Grover, A., Laskin, M., Abbeel, P., Srinivas, A. and Mordatch, I., 2021. Decision transformer: Reinforcement learning via sequence modeling. Advances in neural information processing systems, 34, pp.15084-15097.
>
> > The paper barely discusses the limitations of the proposed approach, like the high computational cost of sampling actions, which might prevent the approach from deployment in some real-world scenarios.
>
> Thank you for your suggestion. We agree it is worth adding a separate section to discuss the limitations in detail, which can now be found in Appendix F of the revised paper.
>
>
> > I think they missed to give details about the optimizer though. But they have provided their code, which I expect (I haven't actually tried) will make the experiments easily and fully reproducible.
>
> Thanks for the catch. We forgot to mention that we used the Adam optimizer for all our training. We have added related descriptions into Appendices B and D, as highlighted in blue.

---

> > ### Comment · Reviewer_J2jU · 2022-11-17
> > **Thanks for the response**
> >
> > I thank the authors for addressing my concerns. I have no further questions. I think this is an interesting paper and I keep my recommendation.

---

### Official Review · Reviewer_jsYy · 2022-11-01

**Confidence:** 4
**Correctness:** 4
**Technical Novelty And Significance:** 4
**Empirical Novelty And Significance:** Not applicable
**Recommendation:** 8

**Clarity, Quality, Novelty And Reproducibility:**

Paper was very clearly written, proposed method is novel to the best of my knowledge, and I do not see any major reproducibility issues.

**Strength And Weaknesses:**

I liked this paper a lot, the authors presented a simple and straightforward method that yielded incredibly powerful results. Given recent advances in diffusion models, the idea of using a diffusion model for behavior cloning as a regularization term for offline RL seems like quite an obvious approach but is to the best of my knowledge quite novel. The paper itself is very well-written with a clear logical flow and was a joy to read. The authors also did a good job of placing their work in the context of other literature and comparing their work to other similar approaches. Experiments are very thorough and shows clear evidence of improvement compared to other baselines

I especially liked the toy example introduced in Section 5 and the Appendix and how the authors separated the behavior cloning part and the policy improvement part. The example gives clear visualization on how the proposed method is advantageous in dealing with multimodal data compared to other approaches. Though one minor suggestion I have is that I think the toy examples perhaps deserve a separate section in the main text.

**Summary Of The Paper:**

This paper proposes using a conditional diffusion policy for offline RL. The proposed method consists of a diffusion model as a behavior cloning term for policy regularization, and a standard policy improvement term via a Q function. The authors demonstrated the effectiveness of their method on a series of toy examples and the standard D4RL, AntMaze, Adroit, and the kitchen benchmarks.

**Summary Of The Review:**

I think this work is well-presented and adds an important contribution to the community, I recommend its acceptance at this venue.

---

> ### Author Response · Authors · 2022-11-10
> **Response**
>
> We thank reviewer jsYy for providing a positive feedback. We appreciate your suggestion to put the toy examples into a separate section. We struggled to do this given the space constraint but will keep this in mind as we revise the paper.

---

### Author Response · Authors · 2022-11-10
**Response to All**

We thank all the reviewers for the time and expertise they have invested in these reviews and for their positive and constructive feedback. We are encouraged that reviewer **jsYy**, **J2jU** and **JRTd** praised the novelty and effectiveness of our method. Your comments and suggestions have helped us to improve the paper. We provide a response and clarifications below for each reviewer respectively and hope they can address your concerns.

We’ve also updated the paper with a few modifications to address reviewer suggestions and concerns (in blue). Summary of updates below:
* We added a discussion of the limitations in the Appendix F.
* We mentioned using the Adam optimizer in Appendix B.
* We added an ablation study of using Gaussian mixture policy in Appendix G.

---

### Public Comment · ~Yinuo_Zhao1 · 2022-11-17
**Question about the combination with the RL policy**

Although the combination of generative method with RL policy is not a new idea [1,2], this paper added the SOTA generative model with the offline DRL as a novel policy regularizer and achieved better result in D4RL benchmark. I wonder if there are more possibilities to combine diffusion models with online DRL, like that in "Parrot" [1]. Specifically, is it possible to compute the logP(a|s) with diffusion models? If true, we can use the log_probs to compute some surrogate objectives in DRL policy updating.


[1] Avi Singh*, Huihan Liu*, Gaoyue Zhou, Albert Yu, Nicholas Rhinehart, Sergey Levine. Parrot: Data-Driven Behavioral Priors for Reinforcement Learning International Conference on Learning Representations (ICLR), 2021.

[2] Yunzhu Li, Jiaming Song, and Stefano Ermon. Infogail: Interpretable imitation learning from visual demonstrations. In Advances in Neural Information Processing Systems, 2017.

---

> ### Author Response · Authors · 2022-11-18
> **Response**
>
> Thanks for pointing out the potential future work. We are currently developing a Diffusion-QL variant for online RL. As it is an ongoing work and is also beyond the scope of this paper, we are refraining ourselves from further discussing it during the review.
>
> Back to your specific question, with the use of diffusion models, $\log p(a_t^0 | s_t)$ in general does not have an analytic probability density function, due to the neural-network-based non-linear transformation in each diffusion step. However, we notice  $p(a_t^0 | s_t)$ can be formulated into a semi-implicit distribution: the first layer is an analytic Gaussian distribution, whose parameters can be i.i.d. drawn from an implicit distribution consisting of all subsequent reverse diffusion layers. Following the method in [1], for a semi-implicit distribution, we may find lower/upper bounds of  $\log p(a_t^0 | s_t)$ and approximate their values via Monte Carlo estimation.
>
> [1] Yin, Mingzhang and Mingyuan Zhou. "Semi-implicit variational inference." International Conference on Machine Learning. PMLR, 2018.

---

> > ### Public Comment · ~Yinuo_Zhao1 · 2022-11-19
> > **Reply**
> >
> > Thanks for your sincere and detailed reply. Overall, I think this is an interesting and logically structured work.

---

### Decision · Program_Chairs · 2023-01-20

**Decision:**

Accept: poster

**Justification For Why Not Higher Score:**

It is extremely straightforward in that it proposes to use diffusion policies to replace other generative models in offline-RL.

**Justification For Why Not Lower Score:**

Though it is simple, the evaluation has merit and may encourage further work.

**Metareview: Summary, Strengths And Weaknesses:**

The paper proposes the use of diffusion policies for offline-RL. It is extremely straightforward in that it proposes to use diffusion policies trained to balance behavior cloning of the policy under which the data was collected, and a policy improvement derived by backpropogating through Q-value function. The Q-value function is learned in a standard way. There evaluation shows improvement on a variety of RL tasks compared to a variety of baselines. There's a discussion on hyperparameters and an ablation study on a few tasks. Though this paper is very straightforward, the combination deserves study.

All reviewers were positive after the author response. The negatives that remains were some residual surprise on the poor performance of other generative models and the need to do dataset specific hyperparameter tuning, though this is discussed to some extent in the paper.

**Note From Pc:**

if the above contains the word "oral" or "spotlight" please see: "oral" presentation means -> notable-top-5% and "spotlight" means -> notable-top-25%. As stated in our emails, we are disassociating presentation type from AC recommendations